# Small-Footprint Wake Up Word Recognition in Noisy Environments Employing Competing-Words-Based Feature

**Ki-Mu Yoon [1] and Wooil Kim [2],\***

[1]   Mediazen Inc., Gangseo-gu, Seoul 07789, Korea; kmyoon@mediazen.co.kr
[2]   Department of Computer Science and Engineering, Incheon National University, Incheon 22012, Korea
**\***   Correspondence: wikim@inu.ac.kr; Tel.: +82-32-835-8459

**Abstract:** This paper proposes a small-footprint wake-up-word (WUW) recognition system for real noisy environments by employing the competing-words-based feature. Competing-words-based features are generated using a ResNet-based deep neural network with small parameters using the competing-words dataset. The competing-words dataset consists of the most acoustically similar and dissimilar words to the WUW used for our system. The obtained features are used as input to the classification network, which is developed using the convolutional neural network (CNN) model. To obtain sufficient data for training, data augmentation is performed by using a room impulse response filter and adding sound signals of various television shows as background noise, which simulates an actual living room environment. The experimental results demonstrate that the proposed WUW recognition system outperforms the baselines that employ CNN and ResNet models. The proposed system shows 1.31% in equal error rate and 1.40% false rejection rate at a 1.0% false alarm rate, which are 29.57% and 50.00% relative improvements compared to the ResNet system, respectively. The number of parameters used for the proposed system is reduced by 83.53% compared to the ResNet system. These results prove that the proposed system with the competing-words-based feature is highly effective at improving WUW recognition performance in noisy environments with a smaller footprint.

**Keywords:** speech recognition; wake-up word; deep neural network; competing words; feature generation

---

## 1. Introduction

As speech recognition systems use large amount of resources, to minimize computational load, many systems employ wake-up-word (WUW) recognition so that they can be awakened to an active mode once WUW is recognized. In earlier research, a support vector machine (SVM) was used for the WUW recognition system [1]. Because the performance of deep neural network (DNN) systems has proven to be highly effective in many fields, there have been numerous efforts to build DNN-based WUW recognizers in various ways [2–12]. In [2], the bidirectional long short-term memory (BLSTM)-based end-to-end model was used to calculate the post-probability similar to the hybrid system, and the weighted finite-state transducers (WFSTs) were used to generate a confidence score from the calculated post-probability. The work in [3] proposed a WUW recognition system that generates six types of confidence measures through the decoder of a DNN–hidden Markov model (HMM) hybrid structure and classifies WUW and non-WUW using SVM.

Many studies on small-footprint keyword spotting have shown effectiveness by employing different types of deep networks, including convolutional neural networks (CNN) [4], convolution

recurrent networks [5], residual networks (ResNet) [6], and other variations [7–9]. Recently, the attention method was applied to small-footprint keyword recognition [10–12]. Various studies have also been conducted to improve recognition performance in noisy environments [13–16].

In this paper, we propose utilizing competing words in order to improve WUW recognition performance and minimize the model size of the system. A high-level feature was generated using the competing-words dataset and the residual network. The competing-words-based feature was used as an input to the CNN-based network for classification network training. For small-footprint systems, we focused on minimizing the size of the model parameters as well as increasing the recognition accuracy. The proposed system was evaluated using the WUW speech data recorded in actual noisy home environments where the real air conditioner was operating. To obtain the training data, the original clean data were filtered using the room response filters with real sounds added as background noise, which were extracted from various television shows.

Section 2 describes the proposed WUW recognition system, including the competing-words-based feature and the structure of the recognizer. The speech database and the experimental results are presented in Section 3, and the concluding statements are provided in Section 4.

## 2. Proposed WUW Recognition System

The motivation for the competing-words-based feature in this study was that phonetically discriminative features across various vocabularies that are obtained independently from the training database could make the WUW recognition system more robust to the unseen non-WUW inputs. In this work, we used competing words to generate more discriminative high-level features over different words. The proposed system consists of two parts: a feature generation network and a classification network. The details are described in the following sections.

### 2.1. Selection of Competing Words

In our work, the WUW was a single two-word command, such as "Hey Siri" and "Okay Google," which consisted of Korean words with four syllables. For the competing-words-based feature generation, we used the PBW452 database, which contains a set of 452 number of phonetically-balanced isolated Korean words where 45 Korean phonemes occur at approximately the same frequency, including a duration of about 19 h uttered by 71 speakers [17]. The PBW452 database does not include the WUW used for our system. From the PBW452 database, we selected $N_{cw}$ number of words for the competing words in three different ways: (1) most acoustically similar words, (2) most acoustically dissimilar words, and (3) mixed set.

To select the most similar words among the 452 number of PBW452 words, we first constructed acoustic models of the 452 words by concatenating the Korean phone models of HMMs using the PBW452 database and the HTK toolkit [18]. Each HMM represented a monophone that consisted of 3 states with an 8-component Gaussian mixture model (GMM) per state. A conventional mel-frequency cepstral coefficient (MFCC) front-end was employed for the input feature to the HMM. Using the acoustic scores of the WUW training data $X$ for the 452 word models, $N_{cw}$ number of words generating the highest scores were selected among the 452 words with the following equation:

$$p(X|w_1^{sim}) > p(X|w_2^{sim}) > \cdots > p(X|w_{N_{cw}}^{sim}) \tag{1}$$

where $w_k^{sim}$ represents the $k$th most similar word to the WUW among the 452 words. The selected competing words $\{w_1^{sim}, w_2^{sim}, \ldots, w_{N_{cw}}^{sim}\}$ were expected to generate an acoustic score relatively similar to that of WUW, such that they might degrade discrimination ability from similar inputs to the actual WUW, resulting in an increase in the false rejection rate (FRR). We also selected the most dissimilar words $\{w_1^{dis}, w_2^{dis}, \ldots, w_{N_{cw}}^{dis}\}$ using the following equation:

$$p(X|w_1^{dis}) < p(X|w_2^{dis}) < \cdots < p(X|w_{N_{cw}}^{dis}) \tag{2}$$

The mixed set of competing words $\left\{w_1^{sim}, \ldots, w_{N_{cw}/2}^{sim}, w_1^{dis}, \ldots, w_{N_{cw}/2}^{dis}\right\}$ also consisted of the same $N_{cw}$ number of words, which were obtained from the top $N_{cw}/2$ words from each of the sets of the most similar words and dissimilar words.

### 2.2. Generation of Competing-Words-Based Feature

As a feature generation network, the ResNet was modified, which showed considerable effectiveness for the small-footprint keyword recognition system proposed by Tang and Lin [6]. The deep neural networks used in our experiments were implemented using the Tensorflow [19]. The proposed feature generation network is presented in Figure 1a. The parameters used for the feature network are listed in Table 1. Here, the network included a single original residual block with double layers of convolution layer and batch normalization, and a reduced residual block with a single layer. The kernel size (*l*) and dilation sizes (*d*) used for the convolution layers are also listed in Table 1. The number of feature maps (*n*) was set to 12. Through the last max-pooling and dense layers, the final output was generated using the softmax function.

Here, the sizes of the last max-pooling and dense layers had a principal impact on the size of the feature generation network, because the dense layers had a much larger number of parameters compared to the convolution layers. Since the output of the max-pooling layer was used as the input to the classification network, its size could also have impacted the performance and the size of the classification network. The size of the dense layer should have been identical to $N_{cw}$, which was the number of selected competing words, because the feature generation network was trained using the dataset of the selected competing words. From a series of extensive experiments on changes to the size of the max-pooing layer and $N_{cw}$, considering the computational expense and recognition performance of the proposed small-footprint system, they were set to 240 and 200 respectively.

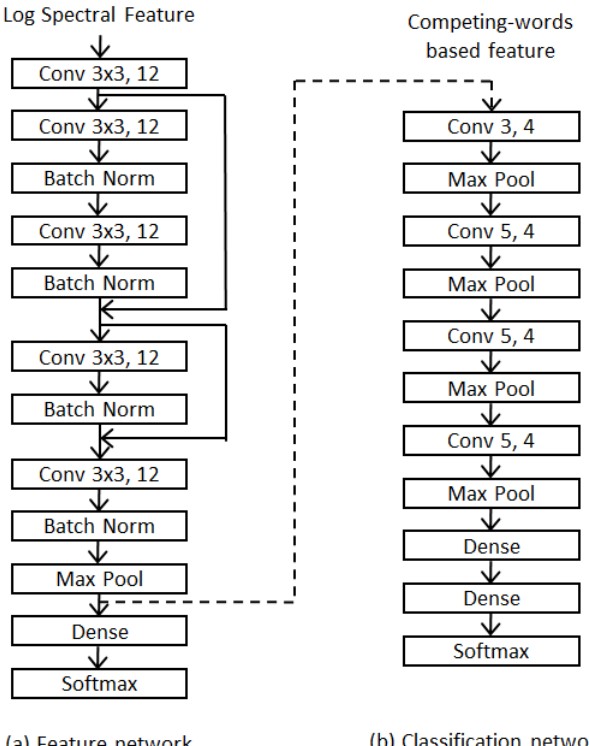

(a) Feature network    (b) Classification network

**Figure 1.** Configurations of the DNN (Deep Neural Networks) for the proposed WUW (wake-up-word) recognition system: (**a**) feature generation network and (**b**) classification network.

**Table 1.** Parameters used for the feature network of the proposed system. * The total number was calculated by excluding the 48,200 parameters of the dense layer, as the dense layer was excluded for the feature generation network of the actual system.

| Type | Kernel (*l*) | Dilation (*d*) | Feature Map (*n*) | Parameters |
|---|---|---|---|---|
| conv | $3 \times 3$ | - | 12 | 108 |
| res $\times 1$ | $3 \times 3$ | $1 \times 1$ | 12 | 2688 |
| conv | $3 \times 3$ | $1 \times 1$ | 12 | 1296 |
| bn | - | - | 12 | 48 |
| conv | $3 \times 3$ | $4 \times 4$ | 12 | 1296 |
| bn | - | - | 12 | 48 |
| max-pool | $23 \times 5$ | - | 12 | 0 |
| dense | - | - | 200 | 48,200 |
| Total | - | - | - | 5484 * |

The input to the feature generation network was a log-spectral feature vector that consisted of 23 elements. The log-spectral feature was the logarithm of the "Mel-filterbank outputs" generated by the ETSI [20] standard using a 25 ms window size and a 10 ms moving interval for 16 KHz sampled speech data. In our work, to extract the high-level feature for an input utterance, 120 frames of log-spectral feature vectors were selected from the center of each speech utterance, and they were added to the feature generation network as an input. Therefore, the input data was two-dimensional as $23 \times 120$.

By applying the six different background noises (i.e., TV programs: drama, news, music, entertainment, sports, and home shopping) at 5 dB to the selected competing words of the PBW452 dataset, including clean conditions, the training data for the feature generation network was constructed. The background noise used for the competing-words training data was identical to the noise samples used for the WUW/non-WUW training database used for the classification network, which are described in more detail in Section 3. A total of 10% of the training data was used as the development data for the training processes in all experiments.

For training the network, each label was encoded as a one-hot vector with 200 elements. The Adam optimizer was used with a learning rate of 0.001, and cross-entropy was used for the loss function. The batch size was set to 1000. The training was stopped when the loss was minimum over the development dataset in all experiments. When the training was finished, the obtained feature network was used for feature generation for classification network training and testing (recognition). The 240-dimensional vector of the output from the last max-pooling layer was used as the input feature vector for the classification network, as shown in Figure 1. Therefore, the number of parameters used for training the feature network was 53,684 in total, including the dense layer. However, only 5484 parameters were used for the feature generation network of the actual system by excluding the dense layer.

*2.3. Classification Network*

For the classification network, a CNN model was employed. The classification network consisted of four blocks of convolution and max-pooling layers. The first convolution layer used 3 as the kernel size (*l*). The other three convolution layers used 5 as the kernel size and 2 as the stride size (*s*), and each max-pooling layer used 2 as the pooling size; consequently, the size of each output element decreased as it went through the layers. In this work, 4 feature maps (*n*) were used. The output size of the last max-pooling layer was 100, and two dense layers were added with 80 and 2 as node numbers. Each convolution layer used the ReLU function, and the first dense layer used the sigmoid function as the activation function. The last dense layer used the softmax function. For training the network, each label was encoded as a one-hot vector. The Adam optimizer was used with a learning rate of 0.001, and cross-entropy was used for the loss function. The batch size was set to 1000.

Table 2 lists the parameters used for the classification network. The total number of parameters was 8510 for the classification network. Therefore, the total number of parameters of the entire system proposed in this paper was 13,994 (= 5484 + 8510). Figure 2 shows a block diagram of the proposed system. In step 1, for the feature network training, the feature network was trained using the competing-words training database. In step 2, using the trained feature generation network, the competing-words-based feature vectors were obtained from the log-spectral feature of the input speech, and then the classification network was trained using WUW and non-WUW data.

**Table 2.** Parameters used for the classification network of the proposed system.

| Type | Kernel ($l$) | Stride ($s$) | Feature Map ($n$) | Parameters |
|---|---|---|---|---|
| conv | 3 | 1 | 4 | 16 |
| max-pool | 2 | 1 | 4 | 0 |
| conv | 5 | 2 | 4 | 84 |
| max-pool | 2 | 1 | 4 | 0 |
| conv | 5 | 2 | 4 | 84 |
| max-pool | 2 | 1 | 4 | 0 |
| conv | 5 | 2 | 4 | 84 |
| max-pool | 2 | 1 | 4 | 0 |
| dense | - | - | 80 | 8080 |
| dense | - | - | 2 | 162 |
| Total | - | - | - | 8510 |

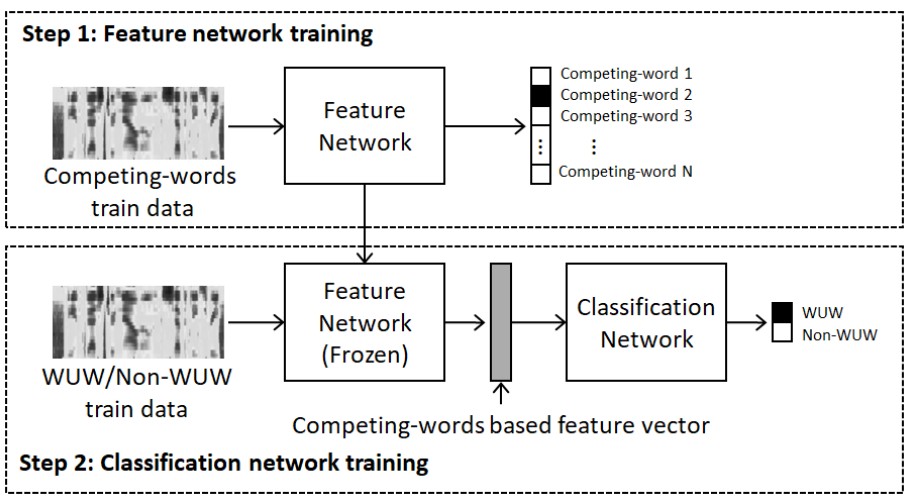

**Figure 2.** Block diagram of the proposed WUW recognition system.

*2.4. Analysis of the Competing-Words Network-Based Feature*

Figures 3 and 4 show the contour curves of principal component analysis (PCA)-transformed vectors of the log-spectral features and the competing-words-based features, respectively. Here, the two main principal components were selected, and they were estimated to be a two-dimensional Gaussian pdf. The contour curves of the distributions were plotted. Figure 3 presents the contour curves of distributions of PCA-transformed log-spectral features for WUW and non-WUW speech samples. Figure 4 presents the contour curves of the PCA transform of the competing-words-based features. In Figure 3, the contours of WUW distribution are almost included in the contours of the non-WUW distribution; therefore, it is difficult to discriminate the WUW distribution from the non-WUW distribution. Compared to Figure 3, the overlapped parts of the contours of WUW and non-WUWs in Figure 4 are considerably smaller. From this observation, we believe that the proposed competing-word-based features obtained discriminative ability between WUW and non-WUWs by

generating from the trained feature network using competing words. The proposed feature could be effective at improving the recognition accuracy of the WUW system.

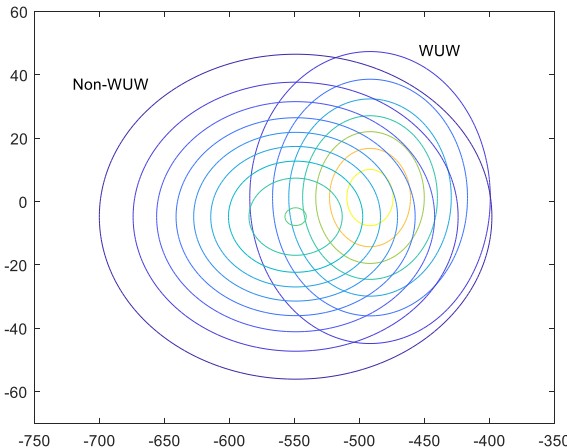

**Figure 3.** Distribution contour curves of two-dimensional vector obtained through PCA (Principal Component Analysis) transform of the log-spectral features for WUW and non-WUW speech utterances.

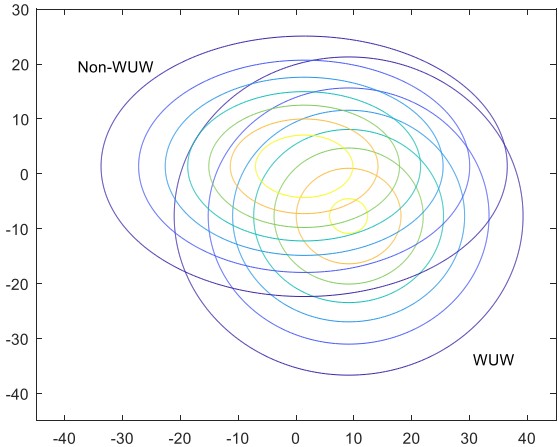

**Figure 4.** Distribution contour curves of two-dimensional vector obtained through PCA transform of the competing-words based features for WUW and non-WUW speech utterances.

## 3. Experiments and Results

### 3.1. Database

In this study, as a WUW, a single Korean two-word command with four syllables was used. The WUW included the name of a Korean company. We obtained 3000 utterances from 1000 speakers who uttered the WUW three times. A total of 2001 utterances from 667 speakers were used for the training set and 999 utterances from 333 speakers were used for evaluation with no overlap in speakers.

The microphone for the input speech to our WUW recognizer developed in this work was assumed to be mounted on an air conditioner in a living room at home. To evaluate performance in the condition most similar to the actual environment where the WUW recognizer was used, the evaluation data was generated by re-recording the evaluation set of the original sound source files (999 samples) through the microphone mounted on the air conditioner with background noise. To simulate the speaker locations from the air conditioner in the actual operation condition, the sound was re-recorded at various locations with a combination of different distances (1 m, 3 m, and 5 m) and angles (left, front, and right), as shown in Figure 5. It was assumed that the air conditioner was in operation while the

user used the speech recognition system, as there was background noise from a wind sound at different levels. In our experiment, 15,984 files were generated for the evaluation dataset, which had a duration of 12.6 h. The datasets used for our experiments are summarized in Table 3.

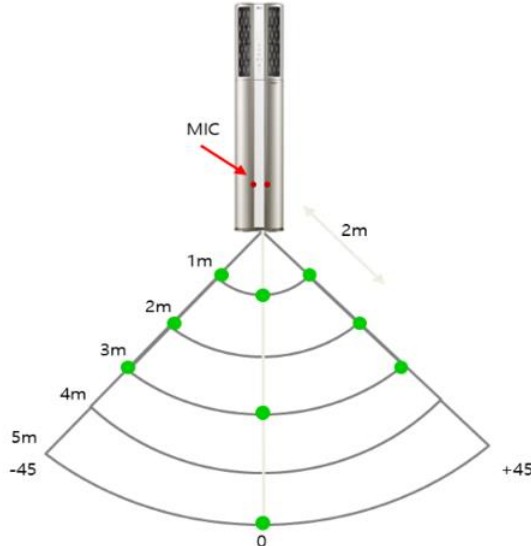

**Figure 5.** Environment where re-recorded utterances were used for testing. Each point represents the position of the speaker. A combination of distances from an air-conditioner and angles were used.

**Table 3.** Datasets used for the experiments. * A total of 10% of each set of training data was used for the development data with no-overlap with the training data.

| Use | Contents | Noise Types | Length (h) |
|---|---|---|---|
| Training/ Development * | Competing-words | Background noise | 6.75 |
| | WUW | RIR + background noise | 9.5 |
| | Non-WUW from TV shows | RIR | 9.5 |
| Evaluation | WUW | Wind noise from air-conditioner | 12.6 |
| | Non-WUW from TV shows | Wind noise from air-conditioner | 45 |

We collected audio data of Korean television shows and programs for evaluation of rejection accuracy for non-WUW utterances that could occur in a home environment. We chose six types of TV programs (drama, news, music, entertainment, sports, and home shopping), and extracted audio data from the Korean TV programs found on YouTube. The audio files with a duration of 45 h were used for the evaluation set, and the audio files with a duration of 9.5 h (i.e., 1.5–2 h for each type of program) were used for the training set with no overlap between evaluation and training. The collected audio data of the TV programs did not include the WUW used in our system.

To simulate a living room environment of an actual house, a room impulse response (RIR) filter was applied to the training datasets. As shown in Figure 6, we generated data by applying the RIR filters with different combinations of microphone positions (1 m, 2.5 m, and 4 m) and reverberation times (0.3 s, 0.4 s, and 0.5 s). For the WUW training data, various types of background noise (white, car, babble, and music) data were added to the generated training data with four different types of signal-to-noise ratios (SNR) (5 dB, 10 dB, 15 dB, and 20 dB). The car and babble noise samples were obtained from NOISEX-92 [21], and the music samples were obtained from Korean pop-music samples. The audio signals collected from the TV programs were also used for the background noise. Through the data generation, we obtained 12,006 files with a duration of 9.5 h for training.

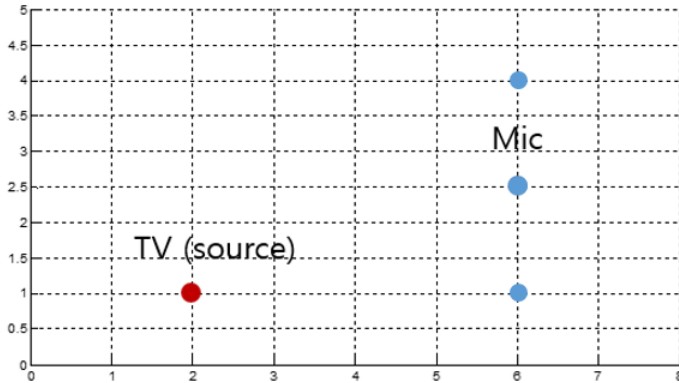

**Figure 6.** Placement of TV and microphone.

### 3.2. Experimental Results

As the performance measure, the equal error rate (EER) and a false rejection rate (FRR) at a 1.0% false alarm rate (FAR) were used, which have been popularly used in literature, including [4–6,10–12]. The FRR and FAR represented the probability of falsely rejecting the WUW inputs and falsely accepting the non-WUW inputs, respectively. The EER was calculated as the point where the FRR and FAR became the same. A value of 100% minus FRR was considered the recognition accuracy of the system for the WUW at a fixed FAR point.

The performance of the proposed method was evaluated according to different sets of the competing words, which were defined in Section 2.1. Here, each system used a 6–7 h long dataset with the selected competing words extracted from the PBW452 database for the feature network training. The results in Table 4 show that the system with a mixed set of competing words showed the best performance in EER and FRR, outperforming the similar and the dissimilar set of competing words. The result of the randomly selected word set showed worse recognition performance in EER compared to the similar set and the worst performance in FFR. From these results, we can see that it is important to properly construct the set of the competing words by including similar as well as dissimilar words to increase the discriminative ability across various inputs.

**Table 4.** Performance comparison of the proposed system for different sets of the competing words in the equal error rate (EER) (%) and in the false rejection rate (FRR) (%) at a 1.0% false alarm rate (FAR).

| Competing-Words Set | EER | FRR |
|---|---|---|
| Most similar | 1.33 | 1.63 |
| Most dissimilar | 1.50 | 1.95 |
| Mixed | 1.31 | 1.40 |
| Randomly selected | 1.45 | 1.98 |

The performance of the proposed method was compared to existing keyword spotting systems that employ CNN [4] and ResNet [6] models, which show effective performance for small-footprint systems and are used as a baseline by many researchers. Tables 5 and 6 list the model structures and parameters used for the baseline systems with the CNN and ResNet models, respectively. The input feature was identical to our proposed system, which was a set of 120 frames of log-spectral features. The same datasets as those of the proposed system were used for training and evaluation. The competing-words data from the PBW452 database was also included in the training data for non-WUW. In the experiments, we tried our best to obtain the smallest size of the neural network models with the best performance in EER and FRR.

**Table 5.** Parameters used for the CNN system.

| Type | Kernel ($l$) | Stride ($s$) | Feature Map ($n$) | Parameters |
|---|---|---|---|---|
| Conv | $3 \times 3$ | $1 \times 1$ | 16 | 160 |
| max-pool | $2 \times 2$ | $2 \times 2$ | 16 | 0 |
| Conv | $3 \times 3$ | $1 \times 1$ | 16 | 2320 |
| max-pool | $2 \times 2$ | $2 \times 2$ | 16 | 0 |
| Dense | - | - | 300 | 537 K |
| Dense | - | - | 100 | 30 K |
| Dense | - | - | 2 | 202 |
| Total | - | - | - | 570 K |

**Table 6.** Parameters used for the ResNet system.

| Type | Kernel ($l$) | Dilation ($d$) | Feature Map ($n$) | Parameters |
|---|---|---|---|---|
| conv | $3 \times 3$ | - | 16 | 144 |
| res $\times 3$ | $3 \times 3$ | $2^{\left\lceil \frac{i}{3} \right\rceil} \times 2^{\left\lceil \frac{i}{3} \right\rceil}$ | 16 | 14,208 |
| conv | $3 \times 3$ | $4 \times 4$ | 16 | 2304 |
| bn | - | - | 16 | 64 |
| avg-pool | - | - | 16 | 0 |
| dense | - | - | 2 | 34 |
| Total | - | - | - | 16,754 |

The CNN model consisted of two layers of 2-D convolution layers and a max-pooling layer with a dense network. The convolution layers had $3 \times 3$ size kernels ($l$) with the ReLU function, and the max-pooling layers had $2 \times 2$ size kernels with $2 \times 2$ strides ($s$). The size of the output of the second max-pooling was 1792. Here, the number of feature maps ($n$) used was 16. The dense network had two hidden layers with 300 and 100 nodes and a sigmoid function. The output layer had two nodes for WUW and non-WUW with a softmax function. The Adam optimizer was used with a learning rate of 0.001, and cross-entropy was used for the loss function. The batch size was set to 1000, which was identical to the proposed system.

The ResNet model was an identical configuration proposed by Tang and Lin [6]. To reduce the size of the model and optimize the performance for our task, that is, a single keyword system, the *res8* model was employed with 16 feature maps ($l$). The ResNet model consisted of the first convolution layer, three residual blocks, and the following convolution layer. As shown in Table 5, 16,754 parameters were used for the ResNet model. The optimizer, loss function, learning rate, and batch size were identical to that of the CNN model.

As another baseline system, we constructed the WUW recognition system proposed in [3], which used six types of confidence measures, and compared the performance to our system. Moreover, we constructed an end-to-end system that concatenated the proposed feature network and classification network in a series with the same number of parameters. The concatenated system is presented as "Concatenated" in Table 7.

**Table 7.** Performance comparison of the baselines and the proposed system in EER (%) and in FRR (%) at 1.0% FAR.

| Systems | EER | FRR |
|---|---|---|
| Ge and Yan [3] | 4.94 | - |
| CNN | 2.55 | 3.93 |
| ResNet | 1.86 | 2.80 |
| Concatenated | 2.79 | 4.45 |
| Proposed | 1.31 | 1.40 |

Table 7 shows that our proposed method outperformed other systems in terms of EER and FRR. In particular, the proposed method showed lower EER compared to the ResNet system, i.e., 1.31 vs. 1.86, respectively. Considering the size of the models in terms of the number of parameters used for the systems (13,994 vs. 16,754, respectively), our system was highly effective for a small footprint device. As for FRR at 1.0% FAR, the CNN and ResNet systems showed 3.93% and 2.80%, respectively, and the proposed system showed 1.40%, which was considerably lower compared to the CNN and ResNet systems. Therefore, the proposed system outperformed the concatenated system as well as the CNN and ResNet systems. These results prove that the proposed two-step system is highly effective at increasing WUW recognition performance, where the feature network is trained over the competing words independently from the training dataset.

Table 8 presents a comparison of the FRR at 1.0% FAR over the different locations of the speaker from the microphone. The farther the speaker moved from the air-conditioner, the smaller the amplitude of the wind noise sound from the air-conditioner became. However, the amplitude also became smaller in amplitude and the input speech became more distorted due to reverberation. From the results in Table 8, the proposed system was consistently robust in the different noisy conditions due to the speaker's location in relation to the air-conditioner compared to the baseline systems. Figure 7 shows the ROC curves of the CNN system, ResNet system, and our proposed system. Here, the marker (*) indicates the point of EER for each system. Table 9 summarizes the numbers of parameters and the number of floating point operations (FLOPs) of the proposed system and baselines systems. The FLOPs were calculated using the "tf.profiler.profile()" function of the Tensorflow.

**Table 8.** Comparison of FRR (%) at 1.0% FAR over the different speaker distances from the microphone.

| Systems | 1 m | 2 m | >3 m | Total |
|---|---|---|---|---|
| CNN | 5.04 | 2.63 | 3.69 | 3.93 |
| ResNet | 3.09 | 3.08 | 2.32 | 2.80 |
| Proposed | 1.15 | 1.53 | 1.57 | 1.40 |

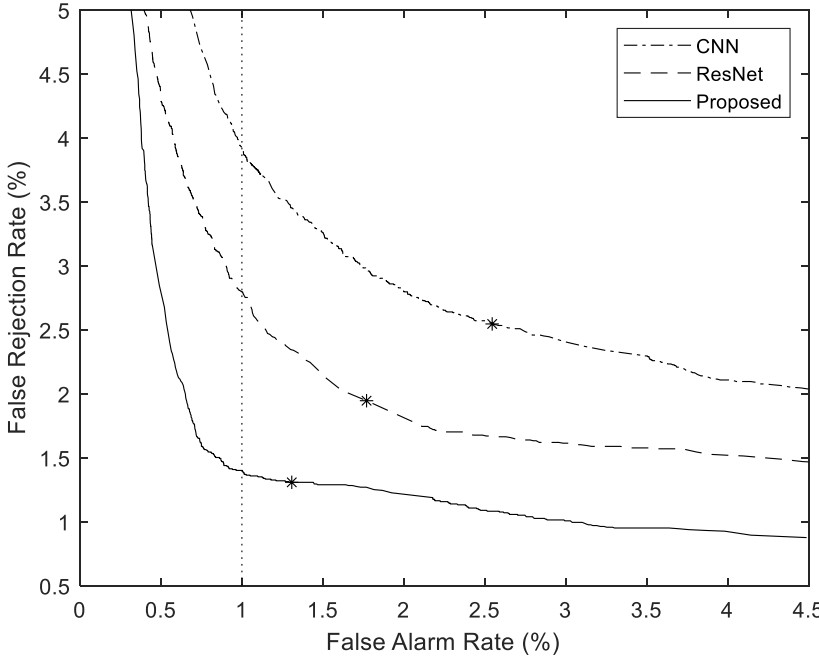

**Figure 7.** ROC (receiver operating characteristic) curves of the baseline systems and the proposed system.

**Table 9.** Summary of the number of parameters and the number of the floating point operations (FLOPs) used in the systems.

| Systems | Parameters | FLOPs |
| --- | --- | --- |
| CNN | 570 K | 1140 K |
| ResNet | 16,754 | 33.5 K |
| Proposed | 13,994 | 28.0 K |

## 4. Conclusions

In this paper, we proposed a small-footprint WUW recognition system for noisy environments by employing the competing-words-based feature. The competing-words-based feature was generated using a ResNet-based DNN, with small parameters using a competing-words dataset. The competing-words dataset consisted of the most similar words and the most dissimilar words to the WUW used for our system. The obtained features were used as the input to the classification network, which was developed using the CNN model. To obtain sufficient data for training, data augmentation was performed by using an RIR filter and adding sound signals from various television shows as background noise, which simulated an actual living room environment. The experimental results demonstrated that the proposed WUW recognition system outperforms the baselines that employ CNN and ResNet models. The proposed system showed 1.31% in EER and 1.40% FRR at 1.0% FAR, which were 29.57% and 50.00% relative improvements compared to the ResNet system, respectively. The number of parameters used for the proposed system was reduced to 83.53% compared to the ResNet system. These results proved that the proposed system with the competing-words-based feature is highly effective at improving the WUW recognition performance in noisy environments with a smaller footprint.

**Author Contributions:** Data curation, K.-m.Y.; Formal analysis, K.-m.Y.; Funding acquisition, W.K.; Investigation, K.-m.Y.; Methodology, K.-m.Y. and W.K.; Project administration, W.K.; Writing—original draft, K.-m.Y.; Writing—review and editing, W.K. All authors have read and agreed to the published version of the manuscript.

**Funding:** This work was supported by an Incheon National University Research Grant in 2016.

**Conflicts of Interest:** The authors declare no conflict of interest.

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
