# Peer review of "Small-Footprint Wake Up Word Recognition in Noisy Environments Employing Competing-Words-Based Feature"

_electronics, doi:10.3390/electronics9122202_

Round 1
Reviewer 1 Report
The paper claims that it presents a small footprint system for the recognition of wake-up-words (i.e. words that are used to trigger speech recognition systems, e.g. "Hey Siri" ). For that reason, the paper presents a deep neural network based system that does the processing, evaluated over some dataset that the paper gathered.
There are several flaws with the paper, specifically:
- The experimental protocol is not presented. That is, there is no presentation at all about the process followed for optimizing the system presented in the paper. For example, how the dataset is used, how the hyper-parameters were optimized, how the model selection was performed, nothing.
- The paper uses a custom dataset, but the description is quite confusing. There is no clarification if only one word is used or many, there is no justification for the data collection decisions, there is no explanation how the training, evaluation, and testing splits were formed.
- The paper says that it presents a small footprint method, but the tables at the results section show that other methods have small amount of parameters.
- In general there are quite many choices for the design of the method and other processes, but no justification of these choices is given.
For the above reasons, I propose the rejection of the paper.
Please find below my detailed comments.
Was there a threshold or minimum limit for p at equation 1? If not, then how the “the selected competing-words are expected to generate an acoustic score similar to that of WUW” is justified? By having no threshold or minimum limit for p means that some p(X|w) could have very low value, basically opposing the claim about “expected to generate an acoustic score similar to that of WUW”. Basically, how the value of 200 was selected?
For equation 2, what was the maximum p value?
Line 88: Why top 100 words?
How the hyper-parameters of the classification network were decided? According to what process and using what data?
Is not clear how the dataset is constructed. There is no mention how many Korean words with four syllabus were used. Then, is not mentioned how the Korean word utterances were split to training and testing splits. Then, it is not mentioned what was the overlap of words between Korean words and TV material, that is how many times was the Korean word(s) found in the TV material? Also, from which room or rooms the IR were used? Why these specific room or rooms?
Line 183: Is not clear what is meant by 45h and 2h.
Line 186: Why is assumed that the WUW recognizer will be mounter on an air conditioner? This is counter-intuitive for both experts and non-experts, given the noise produced by an air conditioner.
Line 196: Which speech recognizer was used? Is there any existing study what the paper replicates? The whole paragraph starting at line 196 is very confusing.
Results: There is no mention or explanation of the experimental procedure. That is, what was the training protocol followed and how the training happened.
Line 215: Why these specific metrics (EER, FRR, and FAR) were used?
Line 259: What data and how these data were used to obtain results for the other methods?
Line 260: What is the point of EER? What does this point show?
Line 262: What is a system developer?
Line 263: What is meant by general use?
Line 263: How the claims of having a general embedding extractor are satisfied by the results? At the moment there are not results backing up these claims.
Line 284: Where is the “small footprint” shown? Tables 5 and 6 show that ResNet has 16754 bout the proposed system has 570 000 parameters. How is that lower that ResNet?
Reviewer 2 Report
In this paper the authors present a new approach to wake up word recognition in noisy environments.
The paper is well organized and well written but some improvements can be done.
The introduction successfully presents the problem and motivations and the state of the art provides a good overview of the technology. Some additional details can be made about competing-words.
The methodology is good but the initial paragraphs, related with the competing-words, can be improved. The process is not very clear and additional details can be provided. The description of the deep-networks is clear but the descriptions of the tables' content can be detailed.
The evaluation shows the good performance of the system but more common metrics could be provided. This can help other authors to compare their systems.
The conclusions make comparisons with other previous works and the references for each case should be provided.
Some detailed suggestions:
L61 "It is well known that the system dependency on the training dataset tends to degrade recognition
performance, especially over the unseen inputs". Maybe this sentence is not the most appropriate, all stochastic based system require an initial training for the adjustment of the involved parameters. This does not create a dependency that degrades or improves performance. The system performance is a consequence of data quality.
L71 This sentence requires additional details. What are "most similar"? Acoustic similarity? Phonetic similarity? "most-dissimilar"? and "mixed"? Criteria? what is "mixed" in particular?
L74 Information about the tool can be interesting for the reader. How were the HMMs trained? Matlab? HTK?
L74 HMM models were also built for the WUW? is X the acoustic representation of WUW?
L74 This paragraph should be re-written. It is not clear what has been done.
L74 w_1^sim represents what?
L91 "the six background noises": What background noises? (they are described below but unknown in this stage of the document)
L91 how do you measure the SNR? is audio amplitude normalized?
Table 1. What is l, d, n, Par?
Table 2. What is l, s, n, Par?
L176 What word? Why this word? You asked 3000 people to record the word?
L271 EER and FRR are useful metrics however when machine learning systems are discusses it is more common to present metrics such as accuracy or F-score. Maybe the authors could also include these as complementary information.
A table showing the score for each noise type or each noise level could also be interesting
L284 What paper is used as baseline for the performance calculations?
Reviewer 3 Report
In this paper, the authors proposed a method based on convolutional neural network for wake-up word recognition in noisy environment. Paper is well written and organized, however there are some issues that must be addressed in the revised version.
Description of the proposed method is not clear. The authors put lots of information together such that the section 2 looks ambiguous. This part needs to be significantly improved.
Although the term “noisy environment” has been mentioned in the title and in the text, there is no clear experiment in the results section showing how noise can changes performance of the algorithm. This should be addressed in the revised version.
Performance indices that have been used to evaluate the method should be defined clearly and reference also needed.
It is suggested that “accuracy” of the method as another performance index will be explored by the authors.
Calculation time for application which is highlighted in the paper is very important. It is suggested that authors add an experiment to the paper for investigating computational time.
It would be great if authors add some examples of false rejection and discuss about the reasons.
Round 2
Reviewer 1 Report
Although the authors addressed many of my comments, the paper still has some problems.
The most prominent one is that the paper does not clearly presents the scientific added value. That is, what existing problem the paper solves and how? How is this problem apparent in the existing approaches, and what are the concrete steps that the paper did in order to fix this problem. If the paper is only about getting better scores without any scientific value, then a conference would be a more suited venue.
The next problem is that is it not clear what is the key idea of the paper. Is the paper about a low footprint system? About the usage of the WUW? The text in the paper totally disregards the footprint aspect until the results section.
Finally, the evaluation conditions seem not to be realistic, since the background noise is missing.
For the above reasons, I propose the major review of the paper. Please find below some detailed comments.
=============================
Line 14: “The competing-words dataset consists of the most similar and dissimilar words to the WUW used for our system” It is not clear what similar and dissimilar means here. Paper must clarify.
Line 24 “The number of parameters used for the proposed system is reduced by 83.53% compared to the ResNet system”: Since ResNet is used for extracting the embeddings of the WUW, and the CNN model for the classification, then why these two are compared? Their functionality is different.
Line 47 “In this paper, we propose to utilize competing-words to improve WUW recognition”: Actually, the title of the paper is about a small footprint network that utilizes the WUW.
Line 60 “ The motivation of this study is that phonetically discriminative features across various vocabulary”: The title of the paper is about a small footprint system, and not about phonetically discriminative features. This sentence is confusing.
Line 69 “phonetically balanced”: is unclear what this means.
Line 75 “HMMs”: this acronym is undefined.
Equation 1 says nothing on how the most similar words were selected. The only thing that Eq. 1 says is that the selected words can be ordered according to their conditionals with X. Thus, the question on how the similar words were selected, remains.
The same goes for Equation 2.
Line 96 “the ResNet is modified”: Modified how?
Line 105 “dominate the size”: What this means? It is unclear.
Line 107 “The size of the dense layer should be identical to ???, which is the number of selected competing-words.” Why? Since the output of the feature generation network is just an embedding, why the size of the embedding should match N_{cw}? This is not correct and is totally unjustified.
Line 109 “From a series of extensive experiments,”: Using what data and how?
Line 117 “Therefore, the input data configures two-dimension “: How the input data can configure something?
Line 136 “For the classification network, the CNN model is employed”: This is unclear. What is the CNN model?
Line 138 “The other three convolution layers use 5 as the kernel size and 2 as the stride size “: This strongly indicate that the employed CNN was 1D CNN. Is this correct?
Line 146 “1,000” Is this one-thousand? Please make clear.
Line 195 “ The microphone for the input speech to our WUW recognizer developed in this work is assumed
to be mounted on an air conditioner in a living room at home. “: Why? Why is assumed to be mounted on an AC? The AC generates lots of noise. To mount a microphone on it, it is counter intuitive. What is the reason for this?
Table 3: Why the training and evaluation conditions are different? Why there is no background noise during evaluation?
Reviewer 3 Report
Paper is acceptable for publishing in this journal.
Author Response
No response to Reviewer 3, since the comment is "Paper is acceptable for publishing in this journal".